# Management Strategies in Arrhythmogenic Cardiomyopathy across the Spectrum of Ventricular Involvement

**DOI:** 10.3390/biomedicines11123259

**Published:** 2023-12-09

**Authors:** Yash Maniar, Nisha A. Gilotra, Paul J. Scheel

**Affiliations:** Division of Cardiology, Department of Medicine, Johns Hopkins University School of Medicine, Baltimore, MD 21287, USA; ymaniar1@jh.edu (Y.M.);

**Keywords:** arrhythmogenic cardiomyopathy, heart failure, arrhythmogenic right ventricular cardiomyopathy, guideline-directed medical therapy

## Abstract

Improved disease recognition through family screening and increased life expectancy with appropriate sudden cardiac death prevention has increased the burden of heart failure in arrhythmogenic cardiomyopathy (ACM). Heart failure management guidelines are well established but primarily focus on left ventricle function. A significant proportion of patients with ACM have predominant or isolated right ventricle (RV) dysfunction. Management of RV dysfunction in ACM lacks evidence but requires special considerations across the spectrum of heart failure regarding the initial diagnosis, subsequent management, monitoring for progression, and end-stage disease management. In this review, we discuss the unique aspects of heart failure management in ACM with a special focus on RV dysfunction.

## 1. Introduction

Arrhythmogenic right ventricular cardiomyopathy (ARVC) was initially described in the 1980s as a hereditary disorder with unique electrophysical properties and fibrofatty replacement of the right ventricle (RV) [1,2,3]. Gene variants mostly in the cardiac desmosome were subsequently implicated [4,5]. Increasing recognition of left ventricle (LV) involvement led to redefinition as “biventricular” ARVC and arrhythmogenic left ventricular cardiomyopathy (ALVC) [6,7,8]. As a spectrum of ventricular involvement was established, these conditions are now described more broadly as arrhythmogenic cardiomyopathy (ACM). However, management and treatment continue to depend on ventricular morphologic phenotypes.

Arrhythmias and sudden cardiac death (SCD) are the hallmark features of ACM but heart failure prevalence is increasing and often under-appreciated [9,10,11]. Increased disease recognition, cascade family screening, and arrhythmic risk stratification have reduced the rate of SCD through appropriate use of implantable cardiac defibrillators (ICDs) and anti-arrhythmic drugs [12,13,14]. With subsequent increased survival, a subset of patients may develop progressive heart failure. Heart failure management guidelines are well established but dictated by LV function [15,16]. Predominant RV involvement in a large proportion of ACM patients leads to unique heart failure considerations in their care. In this review, we discuss different aspects of heart failure management in ACM, focusing on the unique considerations of significant RV involvement.

## 2. Epidemiology and Pathophysiology

Most estimates of prevalence for ACM are based on more narrowly defined ARVC and thought to be about 1:2000 to 1:5000 [17,18], likely an underestimate, with refined terminology and improved recognition. The diagnosis is based on the 2010 revised Task Force Criteria, which leverage distinct RV changes seen on testing in addition to other factors [6,19]. These criteria are less sensitive for non-RV predominant forms. The proportion of patients who will develop heart failure is also unknown but one large cohort study demonstrated 142 of 289 (49%) of patients with ARVC had evidence of heart failure [9].

The diagnosis for all forms is supported by and sometimes predicated on identification of a pathogenic genetic variant but roughly 1/3 of patients meeting ARVC diagnostic criteria are gene-elusive [20], which may extrapolate to LV-dominant phenotypes. Most genetic variants in ACM occur in genes encoding proteins in the cardiac desmosome: plakophilin-2 (*PKP2*, the most common in ACM), plakoglobin (*JUP*, inheritance typically recessive), desmoplakin (*DSP*), desmoglein-2 (*DSG2*), desmocollin-2 (*DSC2*) [21,22]. Variants lead to altered cellular adhesion and signaling through multiple complex mechanisms outside the scope of this review [22]. Several non-desmosomal genes, including desmin (*DES*), integrin-linked kinase (*ILK*), LEM domain containing protein-2 (*LEMD2*), and phospholamban (*PLN*), among others, have also been identified in ACM [22]. The genotype influences phenotype variations. Wooly hair and palmoplantar keratoderma along with ACM are seen with variants in *JUP* (Naxos disease) and recessive forms of *DSP* cardiomyopathy (Carvajal syndrome) [23,24]. *PKP2* variants are associated with more arrhythmias and classical RV-dominant ACM while desmoplakin (*DSP*) and desmoglein-2 (*DSG2*) pathogenic variants are associated with more heart failure [21,25,26,27]. *DSP* pathogenic variants are also associated with an acute myocarditis-like presentation in ACM [28,29]. Non-desmosomal variants, like *DES*, have a wider spectrum of phenotypes including non-ACM cardiomyopathies and are therefore more often identified in LV-dominant forms [22]. Multi-variant patients have an earlier onset of disease for both arrhythmias and heart failure [21]. Conversely, gene-elusive patients seem to have less risk of arrhythmias but a less clear impact on heart failure. Genetic variant significance is based on systematic methodology applied across the genome but these can be difficult to apply to cardiac genetics specifically and should be carried out based on the clinical context in conjunction with expert cardiac genetic counseling [30,31,32]. Variants of unknown significance (VUS), not benign/likely benign or pathogenic/likely pathogenic, are often identified but with time can be reclassified based on familial clustering or identification in other individuals, necessitating periodic variant updating [33,34].

Fibrofatty replacement of the myocardium is the hallmark pathologic finding in ACM as shown in Figure 1. Mechanisms leading to fibrofatty replacement are outside the scope of this review but are actively under investigation and include altered inflammatory signaling due to intrinsic and extrinsic factors [35,36]. This replacement with non-contractile tissue impairs force generation, lessening cardiac reserve and ultimately resting cardiac output, cardinal features of heart failure. Even in patients with RV-dominant forms of ACM, fibrofatty replacement can be seen in the LV and is associated with more arrhythmias and heart failure [1]. A ventricular arrhythmia and SCD are the prototypical presentations of ACM with heart failure often manifesting many years later, sometimes after years of arrhythmia quiescence [9,37,38]. A subset of patients present with heart failure, even advanced disease, without arrhythmias playing a significant role [9]. Finally, some patients will be identified as gene-variant carriers through cascade screening and either be diagnosed with asymptomatic disease or monitored for progression to disease.

## 3. Heart Failure Diagnosis and Assessment

Heart failure is broadly defined as a clinical syndrome with characteristic signs or symptoms caused by a structural and/or functional cardiac abnormality, corroborated by laboratory, imaging, or hemodynamic abnormalities [39]. Heart failure in ACM often manifests as a predominant right-sided heart failure syndrome, though biventricular and LV syndromes are also seen. Left- and right-sided heart failure share many signs and symptoms: lower extremity swelling, unintentional weight gain, jugular venous distension with hepatojugular reflex, exertional dyspnea, and fatigue. Conversely, orthopnea and paroxysmal nocturnal dyspnea (PND) typically indicate LV failure along with exam findings of pulmonary rales and a displaced point of maximal impulse (PMI) as shown in Figure 2. RV failure may have a palpable RV heave, hepatomegaly due to congestion, or a tricuspid regurgitation murmur along with a pulsatile liver. RV failure may have more predominant abdominal distension, anorexia or early satiety from gut edema, or scrotal swelling. In one study of heart failure specifically in ARVC where nearly 50% of patients had clinical heart failure, dyspnea on exertion, fatigue, and abdominal swelling were the highest frequency symptoms [9]. 

Not all ACM patients will develop heart failure but gene carriers and affected patients should be regularly assessed for the more subtle signs of RV dysfunction as outlined in Figure 3. Symptom identification is further complicated by recommendations for exercise restriction to prevent or slow progression in a population of patients that were disproportionately very physically active prior to the diagnosis [40]. Substantial exercise restriction limits symptom opportunity and residual conditioning, especially in younger patients, and may compensate for progressive cardiac dysfunction. Therefore, patients may remain asymptomatic until a significant portion of their cardiac reserve is lost and thereby present at a later stage of disease. Serial assessments of cardiac structure and function, as discussed later, aid in identification of progression. Progressive morphologic changes or any symptoms or subtle signs of heart failure warrant referral to a heart failure specialist for evaluation.

Laboratory tests supplement a heart failure diagnosis in ACM and novel markers may assist risk stratification. N-terminal pro-brain natriuretic peptide (NT-proBNP) correlates with RV dysfunction and RV dilation on cardiac magnetic resonance imaging (CMR) as well as with clinical outcomes including heart transplant and death [41,42]. Increased testosterone in males and decreased estradiol in females were associated with adverse cardiovascular outcomes in ACM in one study, offering a possible explanation for observed sex differences [43]. Serum inflammatory markers such as complement system components have been associated with structural changes and heart failure severity in ACM [44,45]. Many other novel serologic biomarkers including soluble ST2 protein, galectin-3, and extracellular matrix metalloproteinases 2 and 9 are under investigation, but to date have limited clinical availability and according to a 2023 study, provide less prognostic value than NT-proBNP [41]. 

Echocardiography is often the first-line imaging test in heart failure, as it can provide a rapid, noninvasive appraisal of both right and left ventricular structure and function. RV regional akinesis or dyskinesis, an aneurysm of the RV free wall, and RV outflow tract dilation are all structural changes included in diagnostic criteria for ARVC [19]. Right atrial size and degree of tricuspid regurgitation along with RV functional measurements including fractional area change (FAC), tricuspid annular plane systolic excursion (TAPSE), peak systolic excursion velocity (S’), RV early diastolic myocardial velocity (e’), and an RV strain analysis can be assessed and are often abnormal in RV-predominant ACM (Figure 2) [46]. The degree of RV dysfunction on an echocardiogram correlates with the likelihood of clinical heart failure in ACM patients [9], and FAC and TAPSE in particular have been shown to predict adverse outcomes [47,48]. Additionally, septal flattening can suggest RV pressure and/or volume overload, and the inferior vena cava (IVC) diameter and respiratory variation assess volume status.

LV dilation and dysfunction are also easily assessed on echocardiography and may be the primary abnormality in some patients with ACM. LV dysfunction is well recognized as a poor prognostic factor in ARVC [7,49] and an LV ejection fraction (LVEF) < 50% in conjunction with RV dysfunction correlates with an increased risk of death or heart transplant [47]. Echocardiography also provides an easily accessible method for serial assessment to monitor progression of both LV and RV morphologic changes, which may precede worsening heart failure symptoms [50,51]. Repeat echocardiography is recommended every 1 to 3 years to monitor these changes (Figure 3) [6]. The main limitation of echocardiography in general is variable image quality but in ACM, this is further compounded by the geometric layout of the RV often hampering an echocardiographic analysis.

Cardiac magnetic resonance (CMR) imaging overcomes image quality and RV visualization limitations in ACM and also characterizes the myocardial tissue identifying fatty replacement and fibrosis [50]. Current diagnostic criteria do not include CMR tissue characterization but do have CMR-specific RV morphologic parameters that can satisfy major and minor criteria [19]. In ACM, patients with heart failure have a lower RV stroke volume index and RV cardiac index on CMR compared to those without heart failure [9]. In the setting of primarily LV dysfunction, late gadolinium enhancement (LGE) patterns can help identify potential etiologies including ACM. In patients presenting with acute myocarditis, a ring-like and septal LGE pattern was more common in patients later found to have a pathogenic or likely pathogenic variant in a desmosomal gene, most commonly *DSP* [29]. CMR is impractical for frequent serial assessment but occasional repeat assessment, especially in those with poor echocardiogram quality, can identify progression of both RV and LV morphologic changes as well as the degree of fibrosis [50,51]. 

Right heart catheterization (RHC) is the clinical gold standard for diagnosing RV failure and assessing severity. RHC also assesses for alternative causes of RV failure like pulmonary hypertension and LV failure if diagnostic uncertainty exists. Additionally, a simultaneous endomyocardial biopsy for a histopathologic analysis can be performed and is included in the diagnostic criteria for ARVC but is not routinely performed. RV failure in the setting of ACM classically manifests as elevated right atrial pressure (RAP) with normal pulmonary artery pressures and pulmonary capillary wedge pressure (PCWP). If there is LV involvement, PCWP may be elevated and additional parameters such as the RAP-to-PCWP ratio or pulmonary artery pulsatility index (PAPI) help identify the degree of primary RV dysfunction [52]. During the RHC, an estimate of cardiac output (CO) can be obtained using thermodilution or the indirect Fick equation. Thermodilution is the preferred method as it does not rely on assumptions like the indirect Fick method even in the setting of significant tricuspid regurgitation often seen in RV-predominant ACM [52]. Importantly, the measured CO using either method is only at rest and does not assess the ability of the heart to adequately increase CO with activity. RHC should be considered in patients with progressive heart failure symptoms or those with laboratory (rising creatinine or abnormal liver parameters) or imaging abnormalities suggesting more severe heart failure than manifested by symptoms (Figure 3).

Cardiopulmonary exercise testing (CPET) has a well-validated role in assessment of heart failure severity [53], which can likely be extrapolated to heart failure in ACM. Peak oxygen uptake (pVO_2_) quantifies exercise capacity and is used as a marker of heart failure severity with pVO_2_ < 14 cc/kg/min indicating a poor prognosis [54]. Measurement of pVO_2_ requires attaining maximal effort measured objectively with the respiratory exchange ratio (RER) but attaining this in ACM is often limited by anti-arrhythmic blunting of the heart rate or exercise hesitancy. Several effort-independent markers of cardiac function like inefficient ventilatory efficiency (minute ventilation to carbon dioxide (Ve/VCO_2_) slope >34) and the presence of exercise oscillatory ventilation correlate with adverse outcomes in heart failure [53,55] and potentially ACM [56]. CPET in ACM appears to be safe based on a 2020 retrospective study of patients with ARVC undergoing CPET demonstrating no ventricular arrhythmias or other adverse events during or immediately after CPET [56]. Interval CPET testing provides information on the trajectory and need for advanced heart failure referral [53]. 

## 4. Heart Failure Management

### 4.1. Medical Management

Medical management of heart failure aims to stop or slow progression and manage fluid balance to alleviate symptoms as outlined in Figure 4. ACM patients with LV dysfunction should be treated with guideline-directed medical therapy (GDMT) per consensus recommendations [15,16]. These include a β-blocker (βB), mineralocorticoid receptor antagonist (MRA), sodium/glucose cotransporter-2 inhibitor (SGTL2i), and angiotensin receptor/neprilysin inhibitor (ARNI). Second-line for patients who cannot tolerate ARNI therapy is an angiotensin converting enzyme inhibitor (ACEi) or angiotensin II receptor block (ARB) therapy. Similar evidence to guide management of patients with isolated or predominant RV dysfunction is lacking and mostly extrapolated from studies of LV dysfunction and smaller studies [12]. 

ACEi or ARB therapy was associated with slower reductions in TAPSE, RV dilation, and ventricular arrhythmias in one retrospective study [57]. Preemptive initiation of ACEi in mouse models of ARVC type 5 (an aggressive autosomal dominant form) led to reductions in myocardial fibrosis and mortality along with less severe reduction in LV dysfunction [58]. Initiation of ACEi or ARB should be considered in patients with symptomatic RV dysfunction, recognizing the subtle symptoms of heart failure in ACM and the benefit of early initiation, which may aid myocardial remodeling. ARNI therapy is an alternative but may be limited by hypotension and currently no clear evidence exists to recommend this preferentially over ACEi/ARB unless LV dysfunction is present. MRA similarly should be considered in symptomatic RV dysfunction due to beneficial remodeling effects seen in LV dysfunction.

SGTL2i therapy is the newest addition to heart failure GDMT and its benefits can likely be extrapolated to RV dysfunction. An added benefit of SGLT2i is the diuretic effect which can help reduce RV preload. In a plakophilin-deficient ACM mouse model, RV preload reduction with diuretics and isosorbide dinitrate prevented RV dilation [59], and use of isosorbide dinitrate (ISDN) can be considered in patients with symptomatic RV dysfunction [12]. Diuretic therapy, typically with furosemide or torsemide, is the primary method of preload reduction and should be used for ACM patients with evidence of volume overload. In general, there is no difference in outcomes between furosemide or torsemide [60]. However, torsemide has increased bioavailability and absorption is less affected by intestinal edema, which is a predominant manifestation of RV failure. Therefore, torsemide can be considered as an alternative to furosemide for initial therapy and should be utilized in the setting of ineffective preload management with furosemide. Adequate preload reduction needs to be balanced against the preload dependence of a failing RV. Initiation and continuation of these medications should be accompanied by reassessment and monitoring of symptoms along with periodic evaluation of kidney and liver function.

In ACM, β-blockers serve a dual purpose with their anti-arrhythmic properties and a theoretical heart failure benefit. Selection of appropriate anti-arrhythmic therapy in ACM is discussed elsewhere [12,14]. Similar to ACEi/ARB and MRA therapy, βB therapy should be considered in symptomatic RV dysfunction due to known benefits of myocardial remodeling seen in LV dysfunction. However, βBs have negative inotropy, which may be more detrimental in significant RV dysfunction compared to LV dysfunction. Beta-blockers should be started at low doses and up-titrated slowly with close monitoring for worsening symptoms. Safety of starting heart-failure-directed βB will also be influenced by medications used for arrhythmias like sotalol, flecainide, and amiodarone, which may already be slowing the heart rate significantly. Sotalol also has βB properties and combination with a heart failure βB (metoprolol, carvedilol, bisoprolol) must be carried out cautiously. As RV dysfunction progresses, the negative inotropic effects of βB may become more harmful, and down titration or discontinuation in conjunction with arrhythmia management should be considered as symptoms may improve with this intervention. However, even if symptoms improve, the need to decrease the dose is a sign of advancing heart failure and the need for further therapies needs to be considered.

### 4.2. Procedural Therapies

Invasive procedures for management of heart failure are potentially available when medical therapy is inadequate (Figure 4). Chronic resynchronization therapy (CRT) benefits symptomatic patients with heart failure, reduced LVEF, and a widened QRS complex [15,16] and these apply to ACM patients meeting the same criteria. The benefit of CRT or other pacemaker-related therapy in isolated RV dysfunction has not been demonstrated. Utility and selection of patients for ventricular tachycardia (VT) ablation are discussed elsewhere [12,14], and data are lacking for heart-failure-related outcomes. However, the risk of hemodynamic decompensation during VT ablation due to repeated arrhythmias, fluid loading, and extended anesthesia needs to be balanced against the benefit and probability of success.

Progressive RV dilation can eventually lead to significant secondary tricuspid regurgitation (TR), which further exacerbates RV dysfunction. Transcatheter tricuspid valve repair for patients with severe symptomatic TR was recently shown to be safe and improve quality of life but did not affect mortality [61]. It is unclear how effective this therapy would be in predominant-RV-affected ACM but may be an option for patients with significant symptoms and appropriate anatomy without other options for heart failure relief. Remote pulmonary artery pressure monitoring using an implantable sensor prevents heart failure hospitalizations [62] and could be an attractive option for assessment and trajectory monitoring of subtle heart failure symptoms in RV-predominant ACM. However, it is unclear how effective they would be as many of these patients have normal pulmonary artery pressures, a rise in which is the primary parameter being measured with these devices.

### 4.3. Advanced Heart Failure Surgical Therapies

Patients with ACM may progress to develop or even present with advanced or end-stage heart failure. Evidence of advanced heart failure is similar across the LV and RV functional spectrum and includes the need for inotrope therapy, persistent symptoms and/or elevated biomarkers, worsening renal or liver function, repeat hospitalizations for heart failure, an increasing diuretic requirement, and low blood pressure [63]. While typically considered a sign of advanced heart failure, a recurrent ventricular arrhythmia is less applicable in ACM where arrhythmias are inherent to the condition and often occur independent of heart failure [37,38]. Similarly, inability to up-titrate GDMT or having to decrease GDMT is another sign of advanced heart failure that may be less applicable in ACM with significant RV dysfunction and must be considered in the context of RV physiology.

Cardiac transplantation is the definitive therapy for ACM, both for arrhythmia and heart failure clinical phenotypes. Transplants for ACM have increased over the last decade due to improved pre-transplant diagnoses and survival from SCD prevention [64]. A refractory ventricular arrhythmia is an indication for a transplant, but most ACM patients are transplanted because of end-stage heart failure [38,65]. Evaluation for transplant involves extensive multidisciplinary evaluation and diagnostic testing to ensure benefits outweigh risks of transplantation and the potential clinical course thereafter. Significant renal dysfunction, whether due to long-standing heart failure or other causes, may require evaluation for dual heart–kidney transplant. More unique to RV-predominant ACM is the potential for advanced liver disease due to long-standing congestion with case reports of dual heart–liver transplants in ACM [66] 

Once waitlisted for a transplant, multiple patient factors including waitlist status, blood type, sex, size, and geography impact wait time and donor organ availability. In the interim, end-organ perfusion needs to be maintained despite poor cardiac function and may require additional inotropic and/or mechanical circulatory support. These modalities require special consideration in ACM [38]. Inotropes can be arrhythmogenic, worsening the predisposition to ventricular arrhythmias inherent in ACM. This should not preclude their use especially if arrhythmias are a remote issue, but initiation requires careful monitoring of arrhythmia burden. Most commonly used mechanical cardiac support devices are designed for isolated LV support; thus, they are ineffective for predominant-RV dysfunction. Percutaneous RV support options include Protek Duo^®^ (LivaNova, London, UK) and Impella RP^®^ (Abiomed, Danvers, MA, USA) but their use in ACM is often complicated by positioning challenges in a severely dilated RV. An extracorporeal membrane oxygenator (ECMO) provides full cardiac support but carries significant morbidity, including vascular and thrombotic complications. Decisions about transplant timing and peri-operative support require multi-disciplinary discussion and both medical and surgical expertise managing this complex physiology [38].

Durable left ventricular assist device (LVAD) therapy is a more readily available alternative to a transplant in advanced heart failure. In carefully selected patients with LV-predominant ACM, LVAD may be a viable option. However, use is limited in patients with significant RV disease as post-operative RV failure is a significant complication. Another consideration is the potential for increased ventricular arrhythmias post-LVAD caused by scarring around the LVAD cannula site or LV remodeling resulting in cannula myocardial irritation.

### 4.4. Emerging Therapies

Current management of ACM only targets disease manifestations (arrhythmia and heart failure) but pre-clinical work is ongoing to target disease mechanisms for treatment [35]. Investigators developed cell culture and small animal (mouse, zebrafish) models of ACM and targeted either cellular/inflammatory signaling pathways or gene expression [35]. Inhibition of nuclear factor of kappa-B (NFκB) and glycogen synthase kinase-3β (GSK3β) in two different animal models (*DSG2*, *JUP*) improved/preserved cardiac function and decreased myocardial fibrosis, respectively [67,68]. Similarly, small-molecule targeting of peroxisome proliferator activated receptor alpha/gamma (PPARα/γ) reduced cardiac fibrosis in mice and maintained regular electrical activity and calcium handling in human-induced pluripotent stem cells [69,70]. Off-target effects limit direct application clinically but improved understanding facilitates further discovery of potential therapeutics. Alternatively, advances in gene editing technology may allow for direct targeting of gene variants. Current techniques rely on introduction of genetic material into cells via viral vectors to incorporate into genomes and more recently genome editing with discoveries like CRISPR-Cas9. Correction of *DSG2*-deficient cardiomyocytes from stem cells via virally introduced normal *DSG2* has been demonstrated [71]. Implementation of genome-editing technology is actively being investigated in many genetic cardiomyopathies including ACM [72].

## 5. Conclusions

Heart failure prevalence in ACM is increasing and often manifests as predominant-RV dysfunction. This creates unique challenges in diagnosing, risk stratifying, and managing heart failure in ACM. Practitioners caring for these patients need to assess the need for modifications to standard heart failure therapy and determine when such therapy is ineffective. For those patients with progressive heart failure, advanced therapy options exist but evaluation and peri-operative management also require unique considerations. With a full understanding of these unique considerations across the spectrum of heart failure in ACM, practitioners should be able to adequately care for these patients. Additionally, care of these patients in structured multi-disciplinary clinics incorporating electrophysiologists, heart failure cardiologists, genetic counselors, and even translational researchers may improve longitudinal patient care and facilitate implementation of more data-driven management recommendations.

## Figures and Tables

**Figure 1 biomedicines-11-03259-f001:**
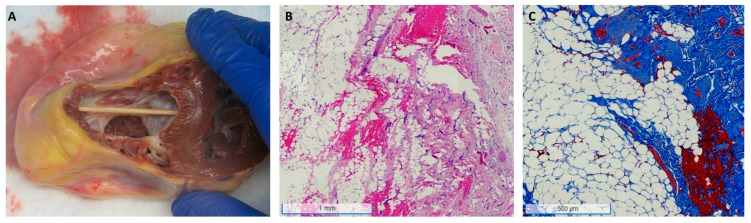
Representative heart explant pathology in arrhythmogenic cardiomyopathy (ACM). (**A**): Gross pathology focusing on the right ventricle demonstrating significant dilation and wall thinning with fatty replacement. (**B**): Hematoxylin and eosin (H&E) stain of right ventricle myocardium at 2×x magnification showing significant fibrofatty replacement surrounding residual cardiomyocytes. (**C**): Masson trichrome stain of the right ventricle at 4×x magnification showing increased fibrosis along with fatty infiltration.

**Figure 2 biomedicines-11-03259-f002:**
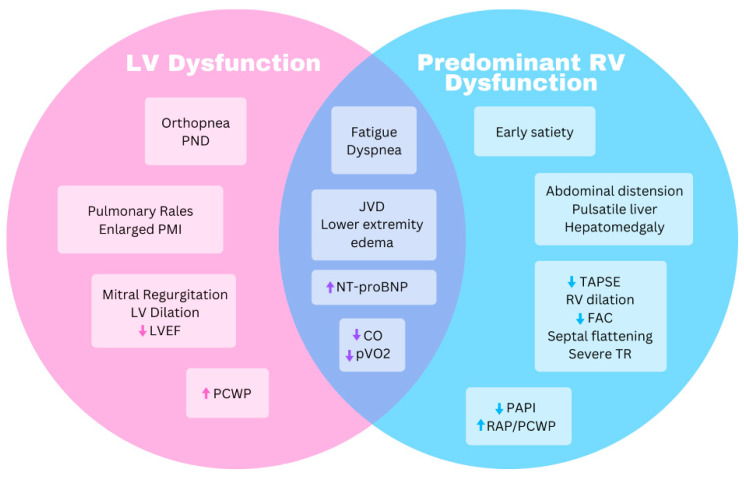
Venn diagram highlighting similarities and differences for symptoms and objective data between predominant right ventricle and left ventricle dysfunction. CO, cardiac output; FAC, fractional area change; JVD, jugular venous distension; LV, left ventricle; LVEF, left ventricle ejection fraction; NT-proBNP, N-terminal prohormone brain natriuretic peptide; PAPI, pulmonary artery pulsatility index; PCWP, pulmonary capillary wedge pressure; PND, paroxysmal nocturnal dyspnea; PMI, point of maximal impulse; pVO_2_, peak oxygen uptake; RAP, right atrial pressure; RV, right ventricle; TAPSE, tricuspid annular plane systolic excursion; TR, tricuspid regurgitation.

**Figure 3 biomedicines-11-03259-f003:**
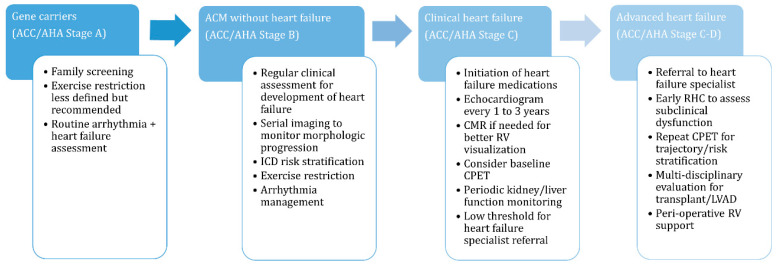
Heart failure management considerations across the arrhythmogenic cardiomyopathy disease spectrum. ACM, arrhythmogenic cardiomyopathy; CMR, cardiac magnetic resonance; CPET, cardiopulmonary exercise test; ICD, implantable cardioverter defibrillator; LVAD, left ventricular assist device; RHC, right heart catheterization; RV, right ventricle.

**Figure 4 biomedicines-11-03259-f004:**
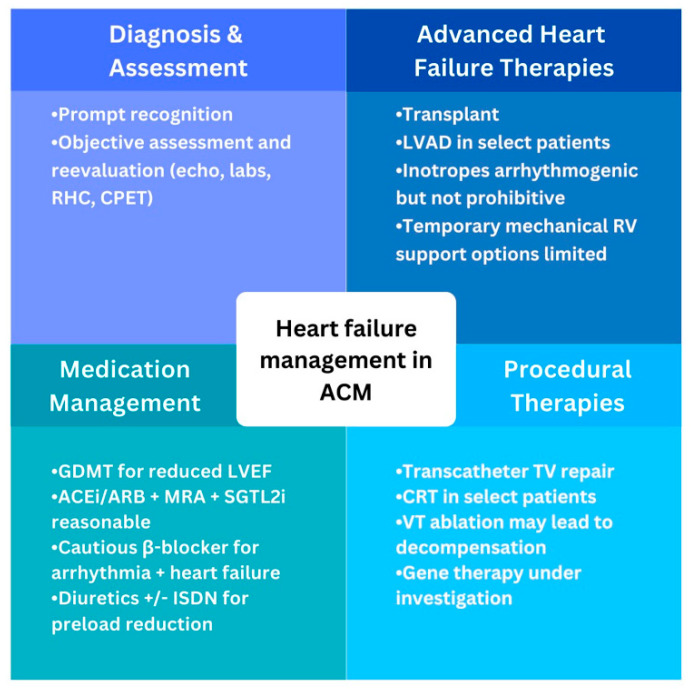
Global overview of heart failure management in arrhythmogenic cardiomyopathy. ACEi, angiotensin converting enzyme inhibitor; ACM, arrhythmogenic cardiomyopathy; ARB, angiotensin receptor blocker; CPET, cardiopulmonary exercise test; CRT, cardiac rexynchronization therapy; GDMT, guideline-directed medical therapy; ISDN, isosorbide dinitrate; LVAD, left ventricular assist device; LVEF, left ventricle ejection fraction; MRA, mineralocorticoid receptor antagonist; RV, right ventricle; SGLT2i, sodium/glucose cotransporter-2 inhibitor; TV, tricuspid valve; VT, ventricular tachycardia.

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
