# Peer review of "Management Strategies in Arrhythmogenic Cardiomyopathy across the Spectrum of Ventricular Involvement"

_biomedicines, 2023, doi:10.3390/biomedicines11123259_

Round 1

Reviewer 1 Report

Comments and Suggestions for Authors

In the review article 'Management Strategies in Arrhythmogenic Cardiomyopathy Across the Sepctrum of Ventricular Involvement' submitted by Maniar et al. to Biomedicines, the authors summarize the clinical presentation of arrhythmogenic cardiomyopathy (ACM). The manuscript is interesting, but needs several extensions and changes:

1.) In the introduction, it would be great if the authors can give a short overview about the desmosomal genes (PKP2, JUP, DSG2, DSC2, DSP) and also about the non-desmosomal genes (DES, ILK, LEMD2, PLN). Maybe the review article 'Insights into Genetics and Pathophysiology of Arrhythmogenic Cardiomyopathy' would be helpful in this context.

2.) Please indicate that DSP and DES mutations are found in patientes with ALVC and PKP2 mutations are more robustly found in classical ARVC patients.

3.) Epidemiology & pathphysiology: Recently, the group of E. van Rooij used spatial transcriptomics to investigate the molecular remodelling in an explanted ARVC heart. In this interesting study, the authors found also severe fibro-fatty replacement in the left ventricle. I would also explain this finding.

4.) In general, it would be great if the authors can explain genetic classification of mutations (beningn, like benign, VUS, likely pathogenic, pathogenic and it would be helpful for the reader, if the authors shortly explain the criteria which are used for this classification. 

5.) I want to ask if the authors can present some images of an typical explanted ACM heart or maybe a MRI picture?

6.)The paragraph about Emering Therapies is too short. Could you please give here first a short overview about animal and cell culture models and than explain gene therapy approaches in more detail. For example, the DSC2 (desmocollin-2) transgenic mice develop ACM caused by fibro-fatty replacement and inflammation. I think in this paragraph it would be worth, if the authors summaryze more relevant animal and cell culture models for ACM. 

In summary, the authors present a nice clinical focussed review . However, at the critized points, the authors can improve their manuscript and could connect the clinical points with more basic resarch studies. In my view this can be optimized in a major revision. Good luck!

Comments on the Quality of English Language

A native speaking editor should double check this manuscript.

Reviewer 2 Report

Comments and Suggestions for Authors

 Authors in this review discuss clinically very serious issue defined as arrhythmogenic cardiomyopathy focusing on dysfunction of the right heart ventricle. Intention is to improve disease recognition and management to prevent heart failure, malignant arrhythmias and sudden cardiac death. Authors emphasize that there is gap of knowledge dealing with dysfunction of the right heart ventricle despite significant proportion of patients suffered from arrhythmogenic cardiomyopathy have predominant or isolated dysfunction of the right ventricle. Article is challenging to pay more attention in guideline directed medical therapy to all aspects of heart failure in arrhythmogenic cardiomyopathy with special attention on right heart dysfunction. As such, gene carriers and affected patients should be regularly assessed for the more subtle signs of right heart ventricle dysfunction.

Article is clearly and comprehensively written with illustration highlighting similarities and differences between predominant right ventricle and left ventricle dysfunction. The most important part is proposal for heart failure prevention and management across the arrhythmogenic cardiomyopathy disease spectrum.

Complex view on arrhythmogenic cardiomyopathy including novel idea for screening, risk stratifying and management may attach interest of both clinicians and researchers for the benefit of patients.

Maybe to conclude article with your imagination how to realize your suggestions in praxis

Round 2

Reviewer 1 Report

Comments and Suggestions for Authors

The authors have improved their manuscript according to the suggested points. I suggest to accept this manuscript for publication.